# Day 10 Post-Prescription Audit Optimizes Antibiotic Therapy in Patients with Bloodstream Infections

**DOI:** 10.3390/antibiotics9080437

**Published:** 2020-07-23

**Authors:** Rita Murri, Claudia Palazzolo, Francesca Giovannenze, Francesco Taccari, Marta Camici, Teresa Spanu, Brunella Posteraro, Maurizio Sanguinetti, Roberto Cauda, Massimo Fantoni

**Affiliations:** 1Dipartimento di Scienze di Laboratorio e Infettivologiche, Fondazione Policlinico Universitario A. Gemelli IRCCS, 00168 Rome, Italy; rita.murri@unicatt.it (R.M.); taccari@hotmail.it (F.T.); teresa.spanu@unicatt.it (T.S.); Maurizio.sanguinetti@unicatt.it (M.S.); roberto.cauda@unicatt.it (R.C.); massimo.fantoni@policlinicogemelli.it (M.F.); 2Dipartimento di Sicurezza e Bioetica, Università Cattolica del Sacro Cuore, 00168 Rome, Italy; claudiapalazzolo84@gmail.com (C.P.); marta.camici@gmail.com (M.C.); 3Istituto Nazionale Malattie Infettive Lazzaro Spallanzani, IRCCS, 00149 Rome, Italy; 4Dipartimento di Scienze biotecnologiche di base, cliniche intensivologiche e perioperatorie, Università Cattolica del Sacro Cuore, 00168 Rome, Italy; brunella.posteraro@unicatt.it; 5Dipartimento di Scienze Gastroenterologiche, Endocrino-Metaboliche e Nefro-Urologiche, Fondazione Policlinico Universitario A. Gemelli IRCCS, 00168 Rome, Italy

**Keywords:** antimicrobial stewardship, bacteremia, medical audit

## Abstract

This study aimed to investigate the clinical and organizational impact of an active re-evaluation (on day 10) of patients on antibiotic treatment diagnosed with bloodstream infections (BSIs). A prospective, single center, pre-post quasi-experimental study was performed. Patients were enrolled at the time of microbial BSI confirmation. In the pre-intervention phase (August 2014–August 2015), clinical status and antibiotic regimen were re-evaluated at day 3. In the intervention phase (January 2016–January 2017), clinical status and antibiotic regimen were re-evaluated at day 3 and day 10. Primary outcomes were rate of optimal therapy, duration of antibiotic therapy, length of hospitalization, and 30-day mortality. A total of 632 patients were enrolled (pre-intervention period, *n* = 303; intervention period, *n* = 329). Average duration of therapy reduced from 18.1 days (standard deviation (SD), 11.4) in the pre-intervention period to 16.8 days (SD, 12.7) in the intervention period (*p* < 0.001). Similarly, average length of hospitalization decreased from 24.1 days (SD, 20.8) to 20.6 days (SD, 17.7) (*p* = 0.001). No inter-group difference was found for the rate of 30-day mortality. In patients with BSI, re-evaluation of clinical status and antibiotic regimen at day 3 and 10 after microbiological diagnosis was correlated with a reduction in the duration of antibiotic therapy and hospital stay. The intervention is simple and has a low impact on overall costs.

## 1. Introduction

Antibiotic-resistant bacteria represent a major and global public health concern [1]. Inappropriate antibiotic use is a major and modifiable cause of antibiotic resistance [2,3]. Data from multiple clinical settings suggest that antimicrobial stewardship programs are associated with improved antimicrobial use [4,5,6]. One of the cornerstones of antimicrobial stewardship programs is that “shorter is better” [7,8]. In this regard, several studies recently showed that for many diseases, a short course of antimicrobial therapy was just as effective as longer courses. Particularly, this was proven not only for urinary tract infections, pneumonias and intra-abdominal infections, but also for bloodstream infections (BSIs) [9,10,11,12,13,14,15,16,17,18,19]. However, at the present moment, the optimal duration of BSI treatment is controversial and yet to be defined. Despite Infectious Diseases Society of America (IDSA) guidelines for the management of intravascular catheter-related uncomplicated BSIs due to Gram-negative bacilli recommending 7–14 days of therapy, several studies have reported similar clinical outcomes for “short course” (≤10 days) and “long course” (>10 days) antibiotic treatments [14,15,16,17,18,19]. Moreover, with the exception of catheter-related BSIs due to coagulase-negative Staphylococci, data on short course treatment for Gram-positive BSIs are still lacking [20,21,22].

Post-prescription audit has been proposed as an efficacious strategy to optimize antibiotic regimens, especially treatment duration. In an earlier report, the present authors demonstrated that a three-day active re-evaluation of all patients with BSI was correlated with a decrease in both antimicrobial therapy duration and hospital stay [23]. The evaluation at the third day, when the antibiogram of the isolated microorganism is acquired, allows us to de-escalate or discontinue useless antibiotic therapy. The optimal duration of antibiotic therapy is unknown. Moreover, previous studies demonstrated that a 10-day course could be sufficient to optimally treat patients with BSIs [14,15,16,17,18,19]. Therefore, the aim of the present study was to determine whether active day 3 and day 10 re-evaluation of patients with Gram-positive or Gram-negative BSIs leads to a further reduction in the duration of antibiotic therapy and hospital stay.

## 2. Results

Between August 2014 and January 2017, 632 patients were enrolled (pre-Iintervention Phase, *n* = 303; Intervention Phase, *n* = 329). The intervention flow chart is shown in Figure 1.

### 2.1. Baseline Characteristics

The baseline demographic and clinical characteristics of the cohort are shown in Table 1. The mean age of the cohort was 71 years (standard deviation (SD), 15.2). In total, 385 (60.9%) of the cohort were males, and 424 (67.1%) of BSI cases concerned patients on medical wards. In total, 224 (35%) patients had been hospitalized within the preceding 90 days, and 150 (24%) had been prescribed an antibiotic within the previous 30 days. In total, 238 (38%) of patients had a central venous catheter in place at the time of BSI diagnosis.

### 2.2. BSI Etiology

BSI etiology across the cohort is shown in Table 2. In total, 104 (16.4%) of the BSI cases were polymicrobial. Of the 528 patients with mono-microbial BSI, a Gram-positive pathogen was isolated from blood cultures in 199 cases (37.7%). Of the 94 BSI cases that were attributable to *Staphylococcus aureus*, 30 (46.9%) were due to methicillin-resistant *Staphylococcus aureus*. A Gram-negative pathogen was isolated in 252 cases (47.7%). Of these, 27 (5.1%) were carbapenemase-producing *Klebsiella pneumoniae*. In 77 cases (12.2%), *Candida* spp. were isolated from blood cultures.

### 2.3. IDS Actions and Inter-Group Differences in IDS Actions at Day 3

In the pre-IP, the re-evaluation at day 3 was feasible in 234 patients (77.2%): 165 patients (54.5%) de-escalated or partially discontinued antibiotic therapy, and 69 (22.8%) were prescribed an extension of antibiotic therapy. In 69 patients (22.8%), the intervention was not feasible: in 31 cases (10.2%) patients were too ill, in 38 cases (12.5%) the therapy was already optimal. In the IP, an intervention at day 3 was feasible for 200 patients (60.8%): of these, 146 (44.4%) de-escalated or partially discontinued antibiotic therapy, and 54 (16.4%) were prescribed an extension of antibiotic therapy. In 129 patients (39.2%), the intervention was not feasible: in 39 cases (11.8%) patients were too ill, in 90 cases (27.3%) the therapy was already optimal. At day 3, considering only people for which an intervention was feasible, a higher rate of de-escalation/discontinuation was observed in IP than in pre-IP (146/200 (73%) vs. 165/234 (70.5%); Table 3). Non-feasibility of the study intervention was higher in period 2 (39.2%) than in period 1 (22.8%).

### 2.4. IDS Interventions at Day 10

In the IP, 153 patients (46.5%) discontinued the ongoing antibiotic therapy at day 10, while 30 patients (9.1%) prolonged the treatment. Indications for an extension of antibiotic therapy comprised the following: persistence of infection and/or a serious clinical condition (19 patients, 63.3%); infections due to a multi-drug resistant organism (MDRO), *S. aureus* or *Candida* spp. infection (five patients, 16.7%); endocarditis (three patients, 10%); prosthetic joint infection (two patients, 6.7%); and spondylodiscitis (one patient, 3.3%). Ten-day re-evaluation of polymicrobial infections took into account every single microorganism which grew from blood cultures. In 17 patients, the 10-day intervention was not feasible because of death (six patients, 35.3%), early discharge (six patients, 35.3%) or transfer to other hospitals (four patients, 23.5%); one patient (5.9%) was lost to follow-up.

### 2.5. Inter-Group Comparison of Clinical and Organizational Outcomes

In the IP, a statistically significant reduction was observed for time to effective therapy and time to optimal therapy. The total duration of therapy was reduced from 18.1 days (SD, 11.4) in pre-IP to 16.8 days (SD, 12.7) in IP (*p* < 0.001). Similarly, length of hospitalization was reduced from 24.1 days (SD, 20.8) in pre-IP to 20.6 days (SD, 17.7) in IP (*p* = 0.001). No inter-group difference was observed for the rate of 30-day mortality (Table 4).

## 3. Discussion

In the present BSI cohort, re-evaluation of clinical status and antibiotic regimen at both three and 10 days after microbiological diagnosis was correlated to better clinical and organizational outcomes. An increased rate of optimal therapy, shorter time between microbiological diagnosis and the beginning of effective therapy, and reduction in both the duration of antibiotic therapy and length of hospital stay were observed during the intervention period. No significant inter-group difference was found in 30-day mortality. In a previous analysis [23], the present authors found that active re-evaluation at day 3 led to an improvement in the rate of optimal therapy and to a reduction in both therapy duration and length of hospitalization, while survival rates were unchanged. After these results, active day 3 re-evaluation of patients diagnosed with BSI was implemented as routine clinical practice by the infectious diseases consultation unit of our hospital. The addition of a 10-day re-evaluation timepoint was correlated with further improvements in these outcomes. The present study involved post-prescription review with feedback.

At the time of writing, the optimal approach to antibiotic stewardship has yet to be defined. Both restrictive and persuasive strategies have been advocated to improve antimicrobial use in the hospital setting [24,25,26]. However, research suggests that “active strategy-based” antibiotic stewardship programs [27], which define care plans with hospital prescribers, might be associated with superior clinical [28] and health-economic [29] outcomes. Several studies have shown that “post-prescription review with feedback” (PPRF) has a greater positive impact in terms of decreasing antibiotic therapy duration than alternative antibiotic stewardship strategies. Tamma et al. [26] compared outcomes for two commonly used antibiotic stewardship strategies: pre-prescription authorization (PPA) and PPRF. Their data indicated that PPRF may have a greater impact on duration of therapy. Further studies have shown that a daily bedside consultation could enhance clinical and health-economic outcomes [30,31,32]. In a recent meta-analysis, bedside consultations were one of the six fundamental strategies to show a significant beneficial effect on at least one outcome [33].

To our knowledge, the present study is the first to evaluate the impact of a day 10, bedside PPRF on clinical and organizational outcomes. Our data confirm that the PPRF approach had a positive impact on BSI management, leading to a reduction in the duration of antibiotic therapy and in the length of hospital stay. Further research is warranted to evaluate the cost-effectiveness of this day 10 strategy in patients with BSI. A PPRF intervention can be implemented by infectious disease specialists, as in our model, or by other physicians trained in antibiotic prescription appropriateness. This kind of intervention is also simple and reproducible in resource-limited or remote settings. The present data suggest that an indirect advantage of this PPRF approach is a general improvement in BSI management.

A previously-published study found that combining antimicrobial stewardship programs with rapid diagnostic tests, such as MALDI-TOF MS, leads to better clinical outcomes [34]. In BSI patients, pathogen identification is crucial in terms of optimizing antimicrobial therapy, and prompt initiation of appropriate antibiotic therapy is associated with improved patient outcomes and decreased healthcare expenditure [35]. In a pre-post quasi-experimental study by Perez et al. [36], MALDI-TOF MS-based organism identification was integrated with an antimicrobial stewardship intervention in patients with Gram-negative bacteremia. The authors demonstrated a non-significant reduction in mortality and a statistically significant reduction in the length of hospitalization. In a pre-post quasi-experimental study by Huang et al. [37], combined MALDI-TOF and antimicrobial stewardship intervention decreased time to organism identification and improved time to both effective antibiotic therapy and optimal antibiotic therapy. Moreover, univariate analysis showed that length of intensive care unit (ICU) stay and recurrent bacteremia were lower in the intervention group, while multivariate analysis showed that the antimicrobial stewardship intervention was associated with a trend toward reduced mortality (odds ratio, 0.55; *p* = 0.075). Notably, the present study revealed an equally high rate of polymicrobial BSI (16.5%) in both study periods. MALDI-TOF MS detects only the single most prevalent strain, and thus a diagnosis of polymicrobial BSI can only be assigned once the growth culture results are available. Hence, in patients with polymicrobial BSI, time to optimal therapy may be longer than in mono-microbial BSI cases.

In patients with BSI, early initiation of antibiotic therapy has been associated with improved prognosis [38]. In the present study, we found a surprising improvement of “early” outcomes, i.e., time to effective and optimal therapy, in the IP. A possible explanation for this is that implementation of the day 10 intervention led to a global improvement in the management of BSI at the study center, including initial approach to treatment. Other possible reasons may be the lower percentage of patients who had received antibiotics in the previous 30 days and of patients who had a central venous catheter (CVC) in place in the IP compared to the pre-IP.

When possible, reducing the duration of antibiotic therapy is a mainstay of antibiotic stewardship programs [39,40], regardless of the healthcare setting, and has shown to be safe compared to longer treatments in selected patient populations [15]. The present intervention was correlated with a significant reduction of 1.3 mean days in the duration of antibiotic therapy when compared to re-evaluation at day 3 only, with no significant inter-group difference in the rate of 30-day mortality (*p* = 0.12). Considering a mean of 1000 BSIs in our hospital per year, the amount of antibiotics saving per year seems to be remarkable. In a cluster-randomized intervention study conducted in 15 small hospitals, Stenehjem et al. showed how an intensive antimicrobial stewardship program including audit and feedback has led to a reduction in total and broad spectrum antibiotic use during the intervention period [41]. In a previous study, the present authors reported a significant decrease in total duration of antibiotic therapy when an active re-evaluation of the patient with a BSI was performed on the third day after starting treatment. Despite this early active intervention, the mean duration of therapy in the cohort remained high (18 ± 11 days) [23]. Interestingly, several recent studies suggested that a ≤10-day course of therapy could be as effective as a longer course in selected patients with BSIs [14,15,16,17,18,19]. In a recent systematic review and meta-analysis, Baur et al. showed that antimicrobial stewardship programs reduced the incidence of infections and colonization with multi-drug resistant Gram-negative bacteria, extended spectrum β-lactamase-producing Gram-negative bacteria and methicillin-resistant *Staphylococcus aureus*, as well as the incidence of *Clostridioides difficile* infections [42]. Importantly, a significant reduction in the length of hospitalization was observed (from 24.1 to 20.6 days).

The present study has several limitations. First, the study was not randomized, and confounders may not have been completely excluded from the analyses. Second, the single-center design precludes generalization of the results to clinical centers with different patient populations, antibiotic stewardship practices, and rates of antimicrobial resistance. Moreover, although of great relevance for antibiotic stewardship programs, patients in hematological and ICU units were excluded due to the study design. Finally, the study was not designed primarily to demonstrate the efficacy of rapid diagnostic tests in terms of reduced time to effective/optimal therapy, time to IDS consultation, or time to beginning of effective therapy.

## 4. Materials and Methods

### 4.1. Study Design and Setting

We conducted a prospective, single-center, pre-post quasi-experimental study. The study was performed at the 1100-bed University Hospital in Rome, Italy (Fondazione Policlinico Gemelli IRCCS, Università Cattolica del Sacro Cuore), where a bedside infectious disease consultancy unit (Unità di Consulenza Infettivologica, UDCI) has been operating since 2012. The UDCI is staffed by four infectious disease specialists (IDSs) and operates from 8 a.m. to 2 p.m. from Monday to Friday. From 2 p.m. to 8 a.m., and on Sunday, an on-call IDS is available. An alert system for blood cultures is active from Monday to Saturday, whereby the microbiologist immediately informs the duty IDS of any positive blood culture result. Patients were enrolled at the time of notification to the IDS of a positive blood culture result (day 0). If not begun before, i.e., at the time of clinical suspicion and blood culture collection, antimicrobial therapy was started at the time of notification to the IDS.

### 4.2. Study Population

All adult patients newly diagnosed with BSI and evaluated by UDCI staff between August 2014 and January 2017 were included. Patients in hematological and ICU units were excluded for setting-related issues, since the UDCI service is not active in these wards and the blood culture alert system follows a separate workflow (i.e., the microbiology service communicates the detection of a pathogen directly to the ward physician on duty).

### 4.3. Data Collection and Definitions

BSI was defined as one or more blood cultures positive with a known pathogen in the presence of clinical signs and symptoms, and at least two blood cultures positive for the same microorganism taken from blood samples for the following microorganisms: coagulase-negative staphylococci, *Corynebacterium* spp., *Micrococcus* spp., *Bacillus* spp., *Propionibacterium* spp., or other similar non-pathogenic microorganisms. Organisms that were non-susceptible to at least one agent in three or more antimicrobial categories were classified as multi-drug resistant (MDR); organisms that were susceptible to only one class in antimicrobial categories were classified as extensively drug resistant (XDR) [43]. Therapy was defined as effective when isolated microorganisms were susceptible to the regimen in vitro. Therapy was defined as optimal when isolated microorganisms showed susceptibility to the ongoing antibiotic regimen and the regimen entailed the following: 1) the narrowest possible spectrum (taking into account allergies to antibiotics and the need for broader coverage due to concomitant infections); and 2) the best possible pharmacokinetic properties. Baseline demographic, epidemiological and clinical data were collected at enrolment. Specifically, we assessed the severity of clinical presentation at BSI onset using the Acute Physiology and Chronic Health Evaluation (APACHE) II score [44].

### 4.4. Study Periods and Interventions

We compared patients enrolled in the pre-intervention phase (pre-IP) to patients enrolled during the intervention phase (IP). In the pre-IP, from August 2014 to August 2015, all patients enrolled were actively re-evaluated at their bedside 72 h after notification by the microbiology laboratory to IDS (day 3), as part of the aforementioned study [23]. In the IP, from January 2016 to January 2017, all patients enrolled were actively re-evaluated both at day 3 and at day 10 after notification.

Possible actions undertaken at day 3 (i.e., in both study periods) comprised the following: 1) de-escalation, i.e., a shift from an empiric, wide spectrum therapy to a more restricted spectrum therapy on the basis of microbiological results; 2) escalation, i.e., a shift from a restricted spectrum antibiotic to a broader spectrum antibiotic; or 3) partial discontinuation, i.e., a reduction in the number of antibiotics on the basis of microbiological results.

As an intervention (only in the IP), all patients were actively re-evaluated at day 10 and a decision regarding discontinuing or prolonging antibiotic therapy was made. In the latter case, data were collected concerning indications for the extension of antibiotic therapy beyond 10 days.

### 4.5. Microbiological Analysis

Blood samples were inoculated in aerobic and anaerobic Bactec (Becton Dickinson Instrument Systems, Sparks, MD, USA) and Bact/Alert bottles (bioMérieux, Marcy l’Etoile, France) and incubated at 35 °C in the Bactec FX and BacT/Alert VIRTUO automated blood culture (BC) systems, respectively. When growth was detected, species level identification of the infecting pathogens was conducted in BC broths using MALDI-TOF MS (MALDI BioTyper; Bruker Daltonik GmbH, Leipzig, Germany)) or FilmArray Blood Culture Identification (BCID, bioMérieux) testing according to the procedure previously described [45]). All specimens were also processed according to the standard procedure that includes subcultures on selective and non-selective media, identification by MALDI BioTyper and antimicrobial susceptibility testing by the Vitek 2 system (bioMérieux) or by the microdilution broth method using commercial dehydrated 96-well panels manufactured by MERLIN Diagnostica GmbH (Bornheim, Germany). Results were interpreted in accordance with the European Committee on Antimicrobial Susceptibility Testing (EUCAST) clinical breakpoints [46]. The phenotypic detection of extended spectrum beta-lactamases (ESBLs), and carbapenemases in *Enterobacterales* isolates was performed according to the EUCAST guidelines [46]. Multi-drug resistance (MDR) and extensive drug resistance (XDR) were defined according to an international expert proposal by Magiorakos et al. [43].

### 4.6. Outcome Measures

Primary outcomes of the study were: rate of optimal therapy, antibiotic therapy duration, length of hospital stay and 30-day mortality. Secondary outcomes included: time to effective antibiotic therapy (i.e., time between notification of the positive blood culture result and first effective treatment) and time to optimal therapy (i.e., time between notification of the positive blood culture result and start of optimal treatment).

### 4.7. Statistical Analysis

A required sample size of 300 patients was calculated for each study period. This calculation was based on a possible decrease in the duration of antibiotic therapy of at least 10%, an assumed difference of 20% in outcome measures between the intervention period and the pre-intervention period, and a power of 80% (= 0.20) at a two-sided significance level of 5% (= 0.05). For the descriptive analyses, the mean and standard deviation (SD) were calculated for continuous values, and the median and interquartile range were calculated for nonparametric continuous variables. Student’s t-test was used to determine inter-group differences in mean values. The chi-square test was used to determine inter-group differences for discrete variables. ANOVA was used to compare mean values for continuous variables. A value of *p* < 0.05 was considered significant. All statistical analyses were performed using SPSS Statistics (IBM SPSS Statistics v23). All analyses were performed in accordance with the Outbreak Reports and Intervention studies Of Nosocomial infection (ORION) statements [47].

### 4.8. Ethical Approval

The study was approved by the institutional review board of the Fondazione Policlinico Universitario A. Gemelli IRCCS—Università Cattolica del Sacro Cuore. The requirement for written informed consent was waived due to the already implemented 3-day re-evaluation of all patients with BSI and the fact that all data were anonymized prior to analysis. All study procedures were performed in accordance with the principles of the Declaration of Helsinki and its later amendments.

## 5. Conclusions

In conclusion, the present data suggest that re-evaluation of clinical status and therapeutic regimen at day 3 and day 10 after microbiological identification is an effective intervention to optimize antibiotic therapy and to improve clinical and organizational outcomes in patients with BSI. We believe that the simple feasibility of the intervention makes it easily reproducible in many healthcare settings, including resource-limited or remote areas.

Improvement of antimicrobial stewardship programs and interventions remains a key healthcare priority, as it will lead to improved clinical outcomes, optimal resource allocation, and improved control of the spread of MDR bacteria. Further studies are needed, especially randomized clinical trials, to assess the impact of PPRF on the duration of therapy for BSIs.

## Figures and Tables

**Figure 1 antibiotics-09-00437-f001:**
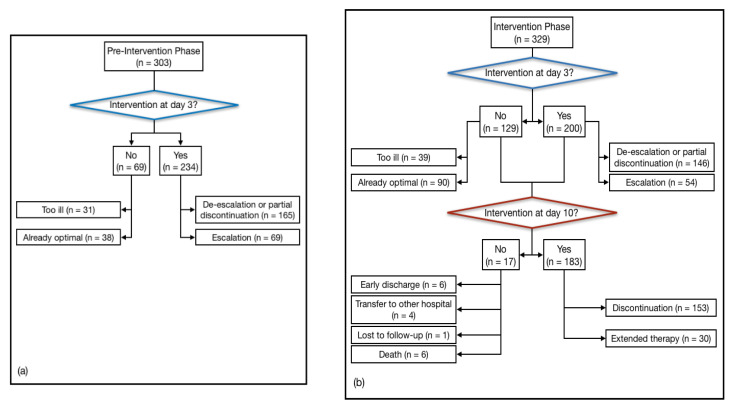
Flow diagram of interventions at day 3 and day 10 in the two study periods. Type and number of interventions implemented in the pre-IP and IP are shown respectively in (**a**) and (**b**).

**Table 1 antibiotics-09-00437-t001:** Baseline demographic and clinical characteristics of the study cohort.

Characteristic	Total *n* = 632	Pre-IP *n* = 303	IP *n* = 329	*p*
Age, mean, years (SD)	71 (15.2)	67.7 (15.6)	69.7 (14.8)	0.15
Males (%)	385 (60.9)	123 (40.6)	124 (37.7)	0.49
Number of comorbidities (SD)	1.00 (0.9)	1.2 (0.9)	0.7 (0.8)	<0.001
Ward (%)				
Medical	424 (67.1)	204 (67.3)	220 (66.9)	
Surgical	208 (32.9)	99 (32.7)	109 (33.1)	0.01
Hospitalization in the previous 90 days (%)	224 (35.4)	116 (38.3)	108 (32.8)	0.08
Antibiotic therapy in the previous 30 days (%)	150 (23.7)	89 (29.4)	61 (18.5)	0.002
Central venous catheter	238 (37.6)	139 (45.9)	99 (30.1)	0.001
APACHE II score, mean (SD)	11.8 (5.9)	12.5 (6.2)	10.2 (5.0)	<0.001
Septic shock (%)	28 (4.4)	14 (5.1)	14 (5.6)	0.95
Polymicrobial (%)	104 (16.5)	66 (21.8)	38 (11.6)	<0.001
Multi-drug resistant BSI (%)	194 (30.7)	100 (33.0)	99 (30.1)	<0.001

APACHE, Acute Physiology and Chronic Health Evaluation; BSI, bloodstream infection; SD: standard deviation. Student’s t-test and Chi-square test were used respectively to compare continuous and categorical variables.

**Table 2 antibiotics-09-00437-t002:** BSI etiology in the study cohort.

Gram-Positive Bacteria	*n* of Samples
Staphylococci	
Coagulase-negative	43
Methicillin-sensitive *Staphylococcus aureus*	64
Methicillin-resistant *Staphylococcus aureus*	30
Enterococci	
*Enterococcus faecalis*	29
*Enterococcus faecium*	9
*Streptococcus pneumoniae*	6
*Streptococcus* spp.	18
**Gram-negative bacteria**	
*Escherichia coli*	69
ESBL-producing	48
*Klebsiella* spp.	26
*K. aerogenes*	
*K. pneumoniae*	26
ESBL-producing	5
Carbapenemase-producing	27
*Acinetobacter XDR^3^*	20
*Proteus mirabilis*	8
MDR	5
*Pseudomonas aeruginosa*	19
MDR	9
*Stenotrophomonas maltophilia*	2

BSI, bloodstream infection; ESBL, extended-spectrum beta lactamases; XDR, extensively drug resistant; MDR, multi-drug resistant.

**Table 3 antibiotics-09-00437-t003:** Interventions at day 3 and day 10.

Intervention	Total *n* = 632	Pre-IP *n* = 303	IP *n* = 329	*p*
Intervention done (%)	434 (68.6)	234 (77.2)	200 (60.8)	<0.001
De-escalation or partial discontinuation (%) at day 3		165 (70.5)	146 (73)	
Escalation (%) at day 3		69 (29.4)	54 (27)	
Discontinuation of therapy after 10 days re-evaluation		-	153 (62.7)	
Study intervention not feasible (%)	198 (31.3)	69 (22.8)	129 (39.2)	<0.001
Patient too ill (%)		31 (44.9)	39 (30.2)	
Optimal therapy already ongoing (%)		38 (55)	90 (69.7)	

Pre-IP, pre-intervention phase; IP, intervention phase. Chi-square test was used respectively to compare pre-IP and IP.

**Table 4 antibiotics-09-00437-t004:** Primary study outcomes.

Outcome	Total *n* = 632	Pre-IP *n* = 303	IP *n* = 329	*p*
Time to start of antibiotic therapy, mean, days (SD)	0.5 (0.9)	0.5 (1.0)	0.49 (0.86)	0.002
Time to start of effective antibiotic therapy, mean, days (SD)	1.0 (1.5)	1.51 (1.8)	0.89 (1.2)	<0.0001
Time to start of optimal antibiotic therapy, mean, days (SD)	2.0 (2.3)	3.03 (2.5)	2.29 (2.1)	<0.0001
Number of effective therapy cases (%)	96.1 (13.2)	89.8 (14.6)	93.3 (11.6)	<0.0001
Number of patients receiving effective therapy (%)	629 (99.5)	301 (99.3)	328 (99.7)	0.01
Number of optimal therapy cases (%)	97 (24.5)	87.2 (28.0)	92.9 (20.5)	<0.0001
Number of patients receiving optimal therapy (%)	589 (93.1)	275 (90.8)	314 (95.4)	<0.0001
Duration of antibiotic therapy, mean, days (SD)	17.40 (14.5)	18.1 (11.4)	16.8 (12.7)	<0.0001
Length of hospitalization, mean, days (SD)	16 (19.3)	24.1 (20.8)	20.6 (17.7)	0.001
Number of deaths at 30 days (%)	105 (16.6)	64 (21.1)	64 (19.5)	0.12

Pre-IP, pre-intervention phase; IP, intervention phase. Student’s t-test and Chi-square test were used respectively to compare continuous and categorical variables.

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
