# Peer review of "Day 10 Post-Prescription Audit Optimizes Antibiotic Therapy in Patients with Bloodstream Infections"

_antibiotics, 2020, doi:10.3390/antibiotics9080437_

Round 1

Reviewer 1 Report

The introduction does not have sufficient background details. Please include the sufficient details and relevant citations

Italicize the microbes names throughout the manuscript

Need to include an ethics statement

There is no mentioning of how the different etiology types impacted the interventions. Were there any differences in the actions undertaken if it was a polymicrobial versus mono case or in an MDR etiology?

Describe how antibiotic stewardship strategy PPRF impacted the study?

The statistics section needs to be verified by a biostatistician for its correctness. ANOVA is theoretically designed to compare the means of continuous distributions. 

Reviewer 2 Report

Introduction: 

The authors may revise the introduction in order to provide a better explanation of the importance of their study. Currently, there are a lot of studies supporting that shorter is better regarding the duration of antibiotic therapy. Authors need to corroborate and comment in order to give a better representation of the current literature.

The clinical reasoning regarding the day 10 selection as the day of the 2nd antibiotic evaluation should be also provided and explained.

Results: In their previous study, authors have included patients up to November 2015. In this study, authors state that the pre-intervention phase started in August 2014. Since there is an overlap between these 2 studies authors should describe the reason for using the previous cohort instead of proceeding with a direct comparison of 3 vs 3 and 10 days. Also in the previous study, group 3 included 313 patients whereas in this study authors include 303 patients.

Lastly, the authors should state why there was no comparison between the other groups of the previous study with this one and whether this was decided a priori?

Discussion:

Authors should describe in more detail and compare with other studies the impact of their results in the antibiotic consumption rate and also in antimicrobial resistance rates.

Authors should include in their limitation paragraph the exclusion of ICU and Hem/Onc patients since these groups have been shown to be of great importance for antimicrobial stewardship programs. 

Conclusion:

A lot of questions remain to be answered regarding the 3 and 10-day re-evaluation of clinical status and therapeutic regimen. Authors should include a paragraph with future studies that need to be performed.

Round 2

Reviewer 1 Report

Proofread the manuscript

Author Response

We spell-checked the whole manuscript and corrected it accordingly. 

Reviewer 2 Report

Dear Authors 

Please find below my comments/suggestion regarding the manuscript.

Tables: Please add as a footnote the statistical test used for the comparison between groups.

Methods: "All analyses were performed in accordance with the ORION statements (http://www.idrn.org/orion.php) on the statistical analysis of intervention studies for nosocomial infections" --> please check that this is indeed the correct website. 

Discussion: Authors state that "In patients with BSI, early initiation of antibiotic therapy has been associated with improved prognosis [38]. In the present study, we found in IP a surprising improvement of “early” outcomes, i.e. time to effective and optimal therapy. A possible explanation is that implementation of the day 10 intervention led to a global improvement in the management of BSI at the study center, including initial approach to treatment."

Another possible explanation would be that patients in the intervention period had received prior antibiotics within 30 days in a lower % and also had less CVC placed than in the Pre-IP.

Also, any changes regarding the ID clinical evaluation after the results of the first study should be discussed in the discussion section.
